

# A comparison of perceptions of nuts between the general public, dietitians, general practitioners, and nurses

Rachel Clare Brown[1], Andrew Robert Gray[2], Lee Ching Yong[1], Alex Chisholm[1], Sook Ling Leong[1] and Siew Ling Tey[1]

[1] Department of Human Nutrition, University of Otago, Dunedin, New Zealand
[2] Biostatistics Unit, Dean's Office, Dunedin School of Medicine, University of Otago, Dunedin, New Zealand

Corresponding author
Rachel Clare Brown,
rachel.brown@otago.ac.nz

## ABSTRACT

**Background.** Nut consumption at the population level remains low despite the well-documented benefits of their consumption, including their cardioprotective effects. Studies have suggested that advice from health professionals may be a means to increase nut consumption levels. Understanding how nuts are perceived by the public and health professionals, along with understanding the public's perceptions of motivators of and deterrents to consuming nuts, may inform the development of initiatives to improve on these low levels of consumption. The aim of this cross-sectional study was to compare perceptions of nuts among three groups of health professionals (dietitians, general practioners, and practice nurses) and the general public in New Zealand (NZ), along with motivators of and deterrents to consuming nuts amongst the general public and their experiences of receiving advice around nut consumption.

**Methods.** The NZ electoral roll was used to identify dietitians, general practitioners (GPs), and practice nurses, based on their free-text occupation descriptions, who were then invited to complete a questionnaire with 318, 292, and 149 respondents respectively. 1,600 members of the general public were randomly selected from the roll with 710 respondents. Analyses were performed using chi-squared tests to look at differences in categorical variables and linear regression for differences in other variables between the four survey groups.

**Results.** Although there were significant differences between the four groups regarding the perceptions of nuts, in general there was agreement that nuts are healthy, high in protein and fat, are filling, and some nuts are high in selenium. We noted frequent agreement that the general public participants would consume more if nuts: improved health (67%), were more affordable (60%), or improved the nutrient content (59%) and balance of fats (58%) within their diets. Over half the respondents reported they would eat more nuts if they were advised to do so by a dietitian or doctor, despite less than 4% reporting they had received such advice. The most frequently selected deterrents to increasing nut consumption were: cost (67%), potential weight gain (66%), and leading to eating too much fat (63%).

**Discussion.** It is concerning that so few among the general public report receiving advice to consume more nuts from health professionals, especially given their apparent responsiveness to such advice. Health professionals could exploit the motivators of nut consumption, while also addressing the deterrents, to promote nut intake. These factors should also be addressed in public health messages to encourage regular nut

consumption among the public. Educational initiatives could also be used to improve the nutritional knowledge of GPs and practice nurses with regard to nuts, although even dietitians were unsure of their knowledge in some cases.

# INTRODUCTION

Nuts have been part of the human diet since paleothic times (*Salas-Salvado, Casas-Agustench & Salas-Huetos, 2011*), providing rich sources of cis-unsaturated fats, fibre, and a number of phytochemicals (*Alasalvar & Bolling, 2015*; *Brufau, Boatella & Rafecas, 2006*; *Ros, 2010*). Although specific types of nuts differ in their micronutrient content, as a food group they can provide substantial amounts of vitamin E, folate, calcium, copper, magnesium, potassium, and selenium to the diet (*O'Neil et al., 2010*; *O'Neil, Nicklas & Fulgoni, 2015*). The regular consumption of nuts is associated with lower total mortality (*Grosso et al., 2015*; *Hshieh et al., 2015*). This reduction is largely due to a lower risk of cardiovascular disease (*Guasch-Ferre et al., 2017*; *Kris-Etherton et al., 1999*; *Nash & Nash, 2008*; *Ros, 2010*; *Ros, Tapsell & Sabate, 2010*; *Sabate, Oda & Ros, 2010*). The regular consumption of nuts has also been associated with reductions in certain cancers including colorectal, endometrial, gastric, lung, and pancreatic (*Aune et al., 2016*; *Bao et al., 2013a*; *Bao et al., 2013b*; *Fadelu et al., 2018*; *Hashemian et al., 2017*; *Lee et al., 2018*; *Lee et al., 2017*; *Van den Brandt & Nieuwenhuis, 2018*; *Wu et al., 2015*). Although nut consumption has been negatively associated with the incidence of diabetes (*Afshin et al., 2014*), these findings are not consistent across studies (*Hshieh et al., 2015*; *Luo et al., 2014*; *Zhou et al., 2014*). A number of epidemiological studies have examined the association between cognitive function and dietary patterns that are characterised by high intakes of nuts, including the Mediterranean and Dietary Approaches to Stop Hypertension (DASH) diets (*Katsiardanis et al., 2013*; *Samieri et al., 2013*; *Valls-Pedret et al., 2012*; *Wengreen et al., 2013*). On the whole, these studies show better cognitive function among those adherent to these dietary patterns. Several mechanisms appear to be implicated in the protective role of nuts, including antioxidant and anti-inflammatory properties, largely related to their nutrient composition, in particular their cis-unsaturated fat, vitamin, and polyphenol content (*Grosso & Estruch, 2016*).

Although the cardioprotective effects of nuts have been widely established, the population intake of nuts has been reported as far from optimal in a number of countries (*Brown et al., 2014*; *Jenab et al., 2006*; *O'Neil et al., 2012*; *O'Neil, Nicklas & Fulgoni, 2015*). For example, in New Zealand (NZ), Europe, and the United States of America (USA) less than 8% of the population on a given day consumed whole nuts (*Brown et al., 2014*; *Jenab et al., 2006*; *O'Neil et al., 2012*; *O'Neil, Nicklas & Fulgoni, 2015*). In addition, the mean population intake of whole nuts is low at less than 3.5. Given that dietary guidelines recommended daily nut intakes of 30 to 42 (*Tey, Brown & Chisholm, 2012*; *US Food and Drug Administration,*

*2003*), it is apparent that strategies to increase regular nut consumption at the population level are needed.

Two studies have suggested that individuals would increase their nut intake if they were advised to do so by their doctor (*Pawlak, Colby & Herring, 2009*; *Pawlak et al., 2012*). Thus, it appears that health professionals may be important facilitators of nut intake. However, despite this willingness to increase nut intake, in both these studies, which included those with or at risk of CVD and/or diabetes (*Pawlak et al., 2012*), or those on a low income (*Pawlak, Colby & Herring, 2009*), there was general agreement among participants that eating nuts on most days of the week would not be consistent with the advice of their doctor. Therefore, it is important to gain an understanding on how health professionals perceive nuts, and how this may influence their advice to consume nuts. Comparison with the general public could assess whether there are any disparities regarding perceptions of nuts which may affect recommendations and nut intakes respectively. In addition, gaining information from the general public on the motivators of and deterrents to nut consumption may provide important information for health professionals to use in practice in order to increase regular nut consumption among their patients and clients.

Therefore, this study aimed to examine the beliefs, attitudes and perceptions of nuts among health professionals and a cross-section of the general public in New Zealand. In addition we aimed to determine the conditional motivators of and deterrents to regular nut consumption among the general population.

## MATERIALS & METHODS

The study methods are described in detail elsewhere (*Brown et al., 2017*; *Yong et al., 2017*), and only essential details are presented here.

### Study design and participants

This was a cross-sectional study, comprising two surveys; one sent to members of the general public and the other to selected health professionals. The general public participants were a national sample of NZ adults aged 18 years and over. Overall, 1,600 participants were randomly selected from the NZ electoral rolls. All NZ citizens and permanent residents aged 18 years or older are required to be registered on the Parliamentary electoral roll. It is estimated that the roll includes 92.6% of all adults (*Electoral Commission, 2014*). Oversampling of Māori (the indigenous people of New Zealand) was used to achieve sufficient numbers for analyses involving this ethnic group.

Health professionals were identified from the NZ electoral roll, which included self-described occupations. We identified individuals whose occupation fell under three main categories: dietitians ($n = 578$), general practitioners (GPs) ($n = 596$), and practice nurses ($n = 266$). These professions reflect those who we considered the most likely to provide dietary advice.

An information sheet was sent to all participants, and completion of the survey was taken as informed consent. The study was approved by the University of Otago Ethics Committee (reference number D14/288).

## Survey development

Two different self-administered questionnaires were developed, one for the general public and one for health professionals. The questionnaires used multiple response questions. Both questionnaires contained questions on socio-demographic characteristics, and assessed perceptions and knowledge of nuts (including nut butter). The general public questionnaire asked about previous advice from health professionals regarding nut consumption, and conditional motivators and deterrents to eating nuts. They were also asked about their intention to eat nuts in the future. The health professional's questionnaire asked the reasons given by their patients for not eating nuts.

Both questionnaires were developed by a group of researchers with expertise in the area of nuts and health, and were pre-tested and modified where appropriate to establish both face and content validity. The general public questionnaire was pretested among a group of 43 members of the general public, whereas the health professional's questionnaire was pretested among six dietitians, two general practitioners, and four practice nurses.

## Survey administration

Both an online version (Survey Gizmo; Widgix Software, LLC, Boulder, Colorado, USA) and a paper copy of the questionnaire were available. A modified version of Dillman's four-stage tailored design method (*Dillman, 2011*) was used for recruitment for both questionnaires. Firstly, an invitation to participate and the URL for the survey were mailed to participants. This was followed seven days later by a postcard thanking those who had already responded and reminding those who had not. After another eight days, a paper version of the questionnaire was sent to the remaining non-respondents. Lastly, a second postcard was sent to recipients of the third mail-out another twelve days later thanking those who had responded and reminding those who had not. Users were provided with an online login code so that each participant could only complete the questionnaire once. Also, each paper questionnaire had a unique code to prevent duplication.

## Statistical analysis

As some respondents to the general public component were health professionals themselves, we removed all those from the sample and respondents who identified as any type of doctor or nurse as well as pharmacists, acupuncturist, dentists, dental nurses and assistants, orthodontists, naturopaths, physiologists, physiotherapists, midwives, veterinarians, and other healthcare workers who would have contact with patients, but not laboratory technicians, or medical secretaries and receptionists.

Appropriate descriptive statistics were calculated for all variables of interest (means and standard deviations for approximately normally-distributed Likert-type scale items and other continuous variables, medians and interquartile ranges for other ordinal and continuous variables, and frequencies and percentages for categorical variables). Chi-squared tests of homogeneity were used to compare the proportion responding "do not know" to each of the 15 belief and perception questions between the four groups (dietitians, GPs, practice nurses, and the general public). Linear regression models were then used to compare the means of the Likert-type responses to the same 15 questions between

**Table 1 Characteristics of survey participants.**

| Demographic | Dietitians | General practitioners | Practice nurses | General population | General population non-health professionals |
|---|---|---|---|---|---|
| n | 318 | 292 | 149 | 710 | 668 |
| Female % (n) | 96.5 (307) | 57.2 (167) | 96.0 (143) | 56.5 (401) | 54.6 (365) |
| Age, years | 42.5 (12.0) | 50.6 (8.4) | 50.9 (10.3) | 52.9 (16.9) | 53.1 (17.1) |
| Ethnicity % (n) | | | | | |
| European | 87.1 (277) | 82.2 (240) | 88.6 (132) | 70.3 (499) | 69.8 (466) |
| Māori | 5.7 (18) | 2.7 (8) | 4.0 (6) | 12.4 (88) | 12.7 (85) |
| Asian | 5.7 (18) | 10.3 (30) | 4.7 (7) | 6.2 (44) | 6.1 (41) |
| Other | 1.6 (5) | 4.8 (14) | 2.7 (4) | 11.1 (79) | 11.4 (76) |
| Employment status % (n) | | | | | |
| Employed | 100 (318) | 100 (292) | 100 (149) | 62.9 (437) | 61.9 (404) |
| Not employed | 0.0 (0) | 0.0 (0) | 0.0 (0) | 37.1 (258) | 38.1 (249) |
| Highest level of education % (n) | | | | | |
| Less than secondary | 0.0 (0) | 0.0 (0) | 0.0 (0) | 2.2 (15) | 2.3 (15) |
| Secondary | 0.0 (0) | 0.0 (0) | 0.0 (0) | 36.6 (252) | 38.4 (248) |
| Post-secondary | 100 (318) | 100 (292) | 100 (149) | 61.2 (421) | 59.3 (383) |
| No. of years as a registered practitioner median (IQR) | 13.0 (20.0) | 20.0 (15.0) | 25.0 (17.5) | – | – |

**Notes.**
All values are means (SD) unless otherwise specified.

the four groups, both without and with adjustment for age, sex, ethnicity, education, and employment. Model diagnostics included inspecting histograms of model residuals for approximate normality and scatter plots of model residuals against fitted values and continuous predictors for a lack of evidence of homoscedasticity or non-linearity. All statistical analyses were performed using Stata 14.2 and, two-sided when available, $p < 0.05$ was considered statistically significant.

# RESULTS

## Response rates
The response rate for the full general population sample was 44% ($n = 710$), and for this sample with the health professionals removed was 43% ($n = 668$). The response rate was 53% ($n = 759$) for the health professionals.

## Participant characteristics
The characteristics of survey respondents are presented in Table 1. The majority of dietitians (97%) and practice nurses (96%) were females, whereas just over half of the general practitioners and general public were females. The mean age of dietitians was around 8 years younger than the other groups, where the mean ages ranged from 51 to 53 years. Most of the participants were NZ European, especially in the health professional

**Table 2  The percentage of general population participants previously receiving advice from a health professional regarding nut consumption ($n = 710$).**

| Previous advice received | Dietitian | General practitioner % ($n$) | Practice nurse | Any of these professionals |
|---|---|---|---|---|
| Advised to eat more nuts or nut butters by | 3.8 (27) | 2.8 (20) | 1.1 (8) | 7.2 (51) |
| Advised to eat less nuts or nut butters by | 0.7 (5) | 1.7 (12) | 0.3 (2) | 2.5 (18) |
| Advised to maintain level of nut consumption by | 0.7 (5) | 1.1 (8) | 0.6 (4) | 2.1 (15) |
| Do not discuss nut consumption with dietitian, GP, or practice nurse | | | | 83.9 (596) |
| Have not seen a dietitian, GP, or practice nurse in the last 5 years | | – | | 6.6 (47) |

groups (ranging from 82% to 89%) compared to the general public (70%). The health professional respondents had a median of 20 years as registered practitioners, with practice nurses having being registered 5 and 12 years longer than GPs and dietitians respectively.

### Previous advice to eat nuts by health professionals

Table 2 shows the percentage of the general population respondents who have previously received advice from a health professional regarding nut consumption. The majority of participants (84%, $n = 596$) reported that they have never received advice from any health professional regarding nut consumption. Only 3.8% ($n = 27$), 2.8% ($n = 20$) and 1.1% ($n = 8$) reported receiving advice to eat more nuts or nut butters from a dietitian, GP, or practice nurse respectively, with 7.2% ($n = 51$) receiving advice from at least one of these groups. Even fewer participants were asked to eat fewer nuts or nut butters, or to maintain their level of consumption by a health professional.

### Intention by the general public to eat nuts in the future

The general public were asked if they would like to increase, decrease, or not change their nut intake. Overall, 73% ($n = 520$) reported they did not want to change the amount they ate, whereas 22% ($n = 159$) reported they would like to increase their nut intake. Only 3% ($n = 24$) reported that they would like to eat fewer nuts.

### Reasons patients give health professionals for not eating nuts

Health professionals were asked to report reasons given by their patients for not eating nuts. The most common reasons were "they are too expensive" (34%, $n = 260$), "they are allergic to nuts" (26%, $n = 200$), "they have dental issues" (22%, $n = 169$), "eating nuts can cause weight gain" (21%, $n = 156$), and "nuts are high in calories" (18%, $n = 130$) with 24% ($n = 184$) indicating either weight gain or calories.

### Conditional motivators to eat more nuts or nut butters by general public

Table 3 shows the responses from the general public regarding potential motivators for eating nuts. The four motivators with the most agreement included "make me healthier", "more affordable", "help me get more nutrients", and "help me get the right balance of

**Table 3  Conditional motivators to eat more nuts or nut butters among the general population (n = 710).**

| Conditional motivators | Strongly agree | Agree | Neither agree nor disagree % (n) | Disagree | Strongly disagree | Mean (95% CI)[*] | Do not know |
|---|---|---|---|---|---|---|---|
| Eating them would help me be healthier | 15.0 (100) | 51.7 (345) | 19.8 (132) | 5.7 (38) | 0.6 (4) | 2.2 (2.1, 2.3) | 7.2 (48) |
| They were more affordable | 26.5 (177) | 33.4 (223) | 21.3 (142) | 11.5 (77) | 2.0 (13) | 2.3 (2.2, 2.3) | 5.3 (35) |
| Eating them would help me get more nutrients | 11.9 (79) | 47.1 (312) | 24.0 (159) | 6.2 (41) | 0.6 (4) | 2.3 (2.2, 2.4) | 10.1 (67) |
| Eating them would help me get the right balance of good fats | 12.1 (80) | 46.2 (306) | 21.6 (143) | 7.1 (47) | 0.6 (4) | 2.3 (2.2, 2.4) | 12.5 (83) |
| A dietitian advised me to | 10.5 (68) | 43.5 (282) | 22.5 (146) | 10.8 (70) | 2.2 (14) | 2.4 (2.4, 2.5) | 10.6 (69) |
| My doctor advised me to | 10.3 (67) | 42.6 (277) | 22.3 (145) | 12.0 (78) | 2.2 (14) | 2.5 (2.4, 2.6) | 10.6 (69) |
| Eating them would help me get the fibre I need | 9.6 (64) | 43.2 (287) | 26.8 (178) | 8.9 (59) | 1.1 (7) | 2.4 (2.3, 2.5) | 10.5 (70) |
| Eating them would help me feel better | 10.0 (66) | 40.5 (268) | 30.4 (201) | 9.4 (62) | 1.1 (7) | 2.5 (2.4, 2.5) | 8.6 (57) |
| Eating them would give me the energy/calories I need | 9.2 (61) | 39.1 (260) | 31.3 (208) | 9.2 (61) | 2.0 (13) | 2.5 (2.4, 2.6) | 9.3 (62) |
| I had more knowledge of recipes involving them | 5.5 (36) | 34.6 (227) | 30.8 (202) | 16.8 (110) | 3.1 (20) | 2.7 (2.7, 2.8) | 9.3 (61) |
| They were lower in fat | 8.7 (57) | 30.2 (198) | 31.0 (203) | 17.1 (112) | 1.7 (11) | 2.7 (2.6, 2.8) | 11.3 (74) |
| They were lower in calories | 8.3 (54) | 27.5 (179) | 32.4 (211) | 18.7 (122) | 1.5 (10) | 2.7 (2.7, 2.8) | 11.7 (76) |
| They were more available where I go shopping | 4.7 (31) | 18.8 (123) | 37.2 (244) | 26.1 (171) | 4.7 (31) | 3.1 (3.0, 3.2) | 8.5 (56) |
| They had more flavor | 3.4 (22) | 14.6 (95) | 33.5 (219) | 33.5 (219) | 8.0 (52) | 3.3 (3.2, 3.4) | 7.0 (46) |

**Notes.**
Values are % (n) except for means and CIs.
[*]1 = strongly agree, 2 = agree, 3 = neither agree nor disagree, 4 = disagree, 5 = strongly disagree.

**Table 4  Conditional deterrents to eating nuts or nut butters among the general public (n = 710).**

| Conditional deterrent | Strongly agree | Agree | Neither agree nor disagree % (n) | Disagree | Strongly disagree | Mean (95% CI)[*] | Do not know |
|---|---|---|---|---|---|---|---|
| If eating them would cost me too much money | 22.1 (146) | 44.9 (297) | 16.0 (106) | 13.3 (88) | 0.6 (4) | 2.2 (2.2, 2.3) | 3.0 (20) |
| If eating them would cause me to gain weight | 20.1 (133) | 45.9 (304) | 17.7 (117) | 10.1 (67) | 1.7 (11) | 2.2 (2.2, 2.3) | 4.7 (31) |
| If eating them would cause me to eat too much fat | 15.1 (99) | 47.9 (314) | 17.7 (116) | 12.2 (80) | 1.2 (8) | 2.3 (2.3, 2.4) | 6.0 (39) |
| If eating them would cause me to eat too many calories | 13.7 (90) | 45.2 (297) | 19.8 (130) | 14.0 (92) | 1.4 (9) | 2.4 (2.3, 2.5) | 5.9 (39) |
| If they are not easily available where I go shopping | 9.4 (62) | 45.4 (298) | 22.2 (146) | 14.8 (97) | 2.1 (14) | 2.5 (2.4, 2.6) | 6.1 (40) |
| If I often forget to eat them even when I have them | 4.9 (32) | 33.2 (218) | 30.3 (199) | 20.9 (137) | 3.5 (23) | 2.8 (2.8, 2.9) | 7.2 (47) |
| If they are difficult to store | 5.2 (34) | 29.8 (194) | 30.1 (196) | 24.7 (161) | 4.8 (31) | 2.9 (2.9, 3.0) | 5.5 (36) |

**Notes.**
Values are % (n) except for means and CIs.
[*]1 = strongly agree, 2 = agree, 3 = neither agree nor disagree, 4 = disagree, 5 = strongly disagree.

fats". Over half of participants agreed or strongly agreed they would eat more nuts if they were advised to so by a dietitian (54%, n = 350) or a doctor (53%, n = 344).

## Conditional deterrents to eating nuts or nut butters

Table 4 shows the responses to questions regarding potential deterrents to eating more nuts or nut butters. The most strongly endorsed deterrent was "if eating them would cost me too much money". This was followed by "if eating them would cause me to gain weight", and "if eating them would cause me to eat too much fat".

**Table 5  Comparison of beliefs and perceptions of nuts between the general public and health professionals.**

| Beliefs and perceptions | Dietitians ($n = 318$) | General practitioners ($n = 292$) | Practice nurses ($n = 149$) | General population Non-health professionals ($n = 668$) | Unadjusted P-value | *Adjusted P-value |
|---|---|---|---|---|---|---|
| Some of them are high in selenium[1] | 1.6 (1.5, 1.6)[a] | 1.9 (1.8, 2.0)[b] | 1.8 (1.7, 1.9)[b] | 2.2 (2.2, 2.3)[c] | <0.001 | <0.001 |
| They are low in energy/calories[2] | 4.5 (4.4, 4.5)[a] | 4.4 (4.3, 4.5)[a] | 4.0 (3.8, 4.2)[b] | 3.5 (3.4, 3.6)[c] | <0.001 | <0.001 |
| They are healthy[1] | 1.7 (1.6, 1.8)[a] | 2.0 (1.9, 2.1)[bc] | 1.9 (1.8, 2.0)[b] | 2.1 (2.0, 2.2)[c] | <0.001 | <0.001 |
| They are high in protein[1] | 1.9 (1.8, 2.0)[a] | 2.0 (1.9, 2.1)[a] | 1.8 (1.8, 1.9)[a] | 2.1 (2.0, 2.1)[b] | 0.001 | 0.013 |
| They are filling[1] | 1.9 (1.9, 2.0)[a] | 2.0 (1.9, 2.1)[ab] | 2.1 (2.0, 2.2)[bc] | 2.2 (2.2, 2.3)[c] | <0.001 | <0.001 |
| They are high in fat[1] | 1.7 (1.6, 1.8)[a] | 2.1 (2.0, 2.2)[b] | 2.1 (2.0, 2.3)[b] | 2.4 (2.1, 2.5)[c] | <0.001 | <0.001 |
| They are low in vitamins & minerals[2] | 4.2 (4.1, 4.3)[a] | 4.1 (4.0, 4.2)[a] | 3.8 (3.6, 4.0)[b] | 3.5 (3.4, 3.6)[c] | <0.001 | <0.001 |
| Eating them can increase people's risk of cardiovascular disease[2] | 4.3 (4.2, 4.4)[a] | 3.8 (3.7, 3.9)[b] | 3.7 (3.6, 3.9)[b] | 3.4 (3.3, 3.5)[c] | <0.001 | <0.001 |
| They are high in antioxidants[1] | 2.1 (2.1, 2.2)[a] | 2.3 (2.2, 2.4)[b] | 2.3 (2.1, 2.4)[b] | 2.5 (2.4, 2.6)[b] | <0.001 | <0.001 |
| They are naturally high in salt/sodium[2] | 4.3 (4.2, 4.3)[a] | 3.7 (3.5, 3.8)[b] | 3.6 (3.4, 3.7)[b] | 3.0 (2.9, 3.1)[c] | <0.001 | <0.001 |
| They are low in fibre[2] | 4.0 (3.9, 4.1)[a] | 3.6 (3.4, 3.7)[b] | 3.6 (3.4, 3.8)[b] | 3.4 (3.3, 3.5)[b] | <0.001 | <0.001 |
| Eating them can increase people's total blood cholesterol[2] | 4.0 (3.9, 4.1)[a] | 3.4 (3.3, 3.5)[b] | 3.4 (3.3, 3.6)[bc] | 3.2 (3.1, 3.3)[c] | <0.001 | <0.001 |
| Some of them are high in iron[4] | 2.8 (2.7, 3.0)[a] | 2.7 (2.5, 2.8)[a] | 2.4 (2.2, 2.6)[b] | 2.3 (2.3, 2.4)[b] | <0.001 | <0.001 |
| Eating them can help lower people's risk of diabetes[3] | 2.6 (2.6, 2.7) | 2.7 (2.6, 2.8) | 2.7 (2.5, 2.9) | 2.8 (2.7, 2.9) | 0.186 | 0.096 |
| Eating them will cause people to gain weight[2] | 3.4 (3.3, 3.5) | 3.2 (3.1, 3.3) | 3.2 (3.1, 3.4) | 3.1 (3.0, 3.2) | 0.017 | 0.068 |

**Notes.**

Values are means (95% Confidence Intervals).

P-values are determined using linear regression.

*Adjusted for age, sex, ethnicity, education, employment.

Values with different superscript letters are significantly different, $P < 0.05$.

1 = strongly agree, 2 = agree, 3 = neither agree nor disagree, 4 = disagree, 5 = strongly disagree.

Note some statements are supported by current evidence and some are worded in contradiction to current evidence.

[1] Statements that are strongly supported by current evidence.

[2] Statements that are strongly contradicted by current evidence.

[3] Statements where current evidence is uncertain.

[4] Some nuts such as pistachios, cashews and almonds contain useful (>4 mg of non-haeme iron/100g) amounts of iron, but bioavailability and significance will rely on other dietary factors.

## Perceptions and knowledge of nuts

Table 5 shows the mean response from the scale of strongly agree = 1 to strongly disagree = 5, for perceptions of nuts for non-health professional general public and health professionals. Although there were statistically significant differences between the four groups regarding the perceptions of nuts (all $P < 0.001$), in general there was agreement that nuts are healthy (mean: 1.7–2.1), high in protein (mean: 1.8–2.1) and fat (mean: 1.7–2.4), are filling (mean: 1.9–2.2), and some nuts are high in selenium (mean: 1.6–2.2). In general, the four groups disagreed with the perceptions that nuts are low in energy (mean: 3.5–4.5), vitamins and minerals (mean: 3.5–4.2), and that eating nuts can increase people's risk of CVD (mean: 3.4–4.3).

There was evidence for a statistically significant difference between groups for all perceptions (all $P \leq 0.013$ for unadjusted and adjusted models), except for "eating them can help lower people's risk of diabetes" and "Eating them will cause people to gain

**Table 6  Comparison of those who responded "do not know" to perceptions of nuts between the general public and three groups of health professionals.**

| Beliefs and perceptions | Dietitians ($n = 318$) | General Practitioners ($n = 292$) | Practice nurses ($n = 149$) | General population non-health professionals ($n = 668$) | P-value |
|---|---|---|---|---|---|
| They are healthy[1] | 0.0 (0)[a] | 0.4 (1)[a] | 0.0 (0)[a] | 6.8 (44)[b] | <0.001 |
| They are filling[1] | 0.0 (0)[a] | 1.7 (5)[b] | 0.7 (1)[ab] | 8.8 (55)[c] | <0.001 |
| They are high in fat1[1] | 0.0 (0)[a] | 0.7 (2)[ab] | 1.4 (2)[b] | 17.4 (107)[c] | <0.001 |
| They are high in protein[1] | 0.0 (0)[a] | 1.4 (4)[b] | 2.0 (3)[b] | 17.8 (110)[c] | <0.001 |
| Eating them will cause people to gain weight[2] | 0.0 (0)[a] | 1.0 (3)[ab] | 1.3 (2)[b] | 20.9 (128)[c] | <0.001 |
| They are low in energy/calories[2] | 0.0 (0)[a] | 0.7 (2)[a] | 0.7 (1)[a] | 22.7 (139)[b] | <0.001 |
| They are low in vitamins & minerals[2] | 0.6 (2)[a] | 4.2 (12)[b] | 2.0 (3)[ab] | 29.0 (182)[c] | <0.001 |
| They are low in fibre[2] | 0.3 (1)[a] | 6.3 (18)[b] | 2.7 (4)[b] | 31.5 (193)[c] | <0.001 |
| They are naturally high in salt/sodium[2] | 0.6 (2)[a] | 10.4 (30)[b] | 6.0 (9)[b] | 31.1 (190)[c] | <0.001 |
| Eating them can increase people's risk of cardiovascular disease[2] | 1.3 (4)[a] | 4.5 (13)[b] | 4.0 (6)[ab] | 39.2 (240)[c] | <0.001 |
| Eating them can increase people's total blood cholesterol[2] | 2.2 (7)[a] | 6.9 (20)[b] | 6.0 (9)[b] | 42.6 (261)[c] | <0.001 |
| They are high in antioxidants[1] | 5.5 (17)[a] | 17.8 (51)[b] | 14.8 (22)[b] | 38.6 (236)[c] | <0.001 |
| Some of them are high in selenium[1] | 2.2 (7)[a] | 18.8 (54)[b] | 18.4 (27)[b] | 44.8 (267)[c] | <0.001 |
| Eating them can help lower people's risk of diabetes[3] | 9.5 (30)[a] | 16.3 (47)[b] | 13.5 (20)[ab] | 53.7 (330)[c] | <0.001 |
| Some of them are high in iron[4] | 10.6 (33)[a] | 25.0 (72)[b] | 17.8 (26)[b] | 41.7 (252)[c] | <0.001 |

**Notes.**

Values are presented as % (number).

P-values were determined using chi-squared test, values with different superscript letters are significantly different, $P < 0.05$.

Note some statements are supported by current evidence and some are worded in contradiction to current evidence.

[1]Statements that are strongly supported by current evidence.

[2]Statements that are strongly contradicted by current evidence.

[3]Statements where current evidence is uncertain.

[4]Some nuts such as pistachios, cashews and almonds contain useful (>4 mg of non-haeme iron/100g) amounts of iron, but bioavailability and significance will rely on other dietary factors.

weight", with the latter significant only in the unadjusted model. For all other questions, the responses of dietitians differed statistically significantly from the general public without and with adjustment for age, sex, ethnicity, education, and employment status.

Table 6 shows the percentage of health professionals and the non-health professional general public ($n = 668$) who responded 'I do not know' to perception statements regarding nuts. The statements which received the highest responses for this were regarding "eating them can help lower people's risk of diabetes" (10–54%), "some of them are high in selenium" (2–45%), "eating them can increase people's total blood cholesterol" (2–43%), "some nuts are high in iron" (2–43%), and "they are high in antioxidants" (6–39%). There was evidence of a statistically significant difference between the four groups (three health professional and the general public) for each of the perceptions (all $P < 0.001$). Overall, dietitians were least likely to indicate a lack of knowledge, whereas the general public were most likely to respond "I do not know". Both GPs and practice nurses tended to be intermediary in their responses.

The statements that health professionals were most unsure of included "some of them are high in iron" (11–25%) and "eating them can help lower people's risk of diabetes" (10–16%). However, as in the general trend for the other perceptions, the percentage of the general public responding "I do not know" to these two statements was statistically significantly higher than the three health professions, and significantly more GPs and practice nurses were likely to respond in this way compared to dietitians.

## DISCUSSION

To the best of our knowledge, this is the first study to investigate motivators of, and deterrents to, nut intake among the general population, and to compare the perceptions of nuts between the general public and health professionals. Very few general public respondents reported receiving advice to eat nuts regularly from health professionals.

In general, there was agreement among the groups for the perceptions that nuts are healthy, high in protein and fat, are filling, and some nuts are high in selenium. There was an overall pattern of higher agreement amongst dietitians, followed by GPs and practice nurses, with the lowest level of agreement among the general public. There appeared to be some confusion among groups regarding the effects of nut consumption on diabetes risk and weight gain.

It is concerning that less than 4% of the general public had previously been advised to eat more nuts by a dietitian, GP or practice nurse. This seems especially low given that there was strong agreement among health professionals that nuts are healthy. Also, it appears that the general public would be responsive to such advice, given that over 50% of participants reported that they would eat more nuts if advised to do so by a dietitian or a doctor. These findings are consistent with those of Pawlak et al., who reported strong agreement among their low-income participants that they would eat nuts if advised to do so by their doctor (*Pawlak, Colby & Herring, 2009*). Also, among a cohort with or at risk of CVD and/or diabetes, 63% of participants claimed they would eat nuts on most days of the week if this was recommended by their doctor (*Pawlak et al., 2012*). In both these studies, participants did not generally agree that eating nuts on most days of the week was consistent with advice from their doctor. Collectively, the results from the present study and those by *Pawlak, Colby & Herring (2009)* and *Pawlak et al. (2012)* suggest that the general public are not receiving advice to eat nuts regularly by health professionals, but many would do so, if they received such advice. Therefore, health professionals are potentially important facilitators of regular nut consumption.

In order to increase population intakes of nuts, it is useful for health professionals to understand what would motivate their patients to eat more nuts, and what might deter them. These barriers and facilitators also provide useful information for designing public health messages aimed at increasing nut intake. The most popular motivators to eat more nuts were largely related to health outcomes e.g.,"would help me feel healthier" and "would help me get more nutrients". This indicates that people are largely interested in their health. Therefore, highlighting the health benefits of eating nuts on a regular basis is important. This information should be included in promotion material encouraging nut consumption.

The top deterrent to eating more nuts was "if they cost too much". This is congruent with the second most prevalent motivator, which indicated people would eat more if they were more affordable. Further to this, 34% of health professionals reported that their patients did not eat nuts because they considered them to be too expensive. Nuts range in price in NZ (1US dollar = 1.48NZ dollar) from around NZ$0.21 (peanuts) to NZ$2.37 (pine nuts) per 30 g serve. The less expensive types are peanuts, almonds, and cashew nuts. Peanut butter is also an affordable form at around NZ$0.36 per 30 g serve. Therefore, these types of nuts, especially in forms with no added sugar, salt, and oil, could be promoted as the most affordable, especially to those on a low income. In addition, it is important to put the price of nuts into context with other snacks where they compare favourably to fruit (e.g., one banana NZ$0.70), cereal bars (around NZ$0.50 per bar), and potato crisps (around NZ$1.49 per 40 g packet).

Other deterrents to nut consumption included if they "cause me to eat too much fat", "cause me to gain weight, and "cause me to eat too many calories". These latter two were also reasons that around one-fifth of patients gave to health professionals for not eating nuts. It therefore appears that some of the general public are worried about weight gain. Nuts are energy-dense and high in fat. However, epidemiological research suggests that regular nut consumers are leaner than non-nut consumers (*Bes-Rastrollo et al., 2007*; *Bes-Rastrollo et al., 2009*; *Martinez-Gonzalez & Bes-Rastrollo, 2011*; *Mozaffarian et al., 2011*). Also, intervention studies have shown that the addition of nuts to the regular diet results in no weight gain, or less weight gain than predicted (based on the extra energy from the nuts) (*Alper & Mattes, 2002*; *Flores-Mateo et al., 2013*; *Hollis & Mattes, 2007*; *Mattes, Kris-Etherton & Foster, 2008*; *Tey et al., 2011*). Suggested mechanisms for this finding include increased satiety (*Brennan et al., 2010*; *Mattes, 2008*; *Tan & Mattes, 2013*) increased metabolic rate (*Alper & Mattes, 2002*; *Claesson et al., 2009*; *Coelho et al., 2006*), and increased loss of metabolisable energy in the faeces (*Ellis et al., 2004*; *Gebauer et al., 2016*; *Grundy et al., 2015*; *Grundy, Lapsley & Ellis, 2016*; *Novotny, Gebauer & Baer, 2012*). Given that the fear of weight gain may be deterring some people from consuming nuts on a regular basis, nut-promoting messages should include information from the wealth of research suggesting that adding nuts to the diet does not result in weight gain, especially when nuts are used to replace other less healthful snacks. In addition, it is likely that there needs to be more education aimed at health professionals in this area, given the majority of these participants seemed unsure about the potential of nuts to cause weight gain. In fact, around 10% of dietitians and around one-fifth of GPs and practice nurses agreed that eating nuts caused weight gain. Thus targeted education on this topic is needed to improve the advice given to patients.

A further barrier to regular nut consumption highlighted by the current research was dentition issues. Over one-quarter of health professionals reported that their patients did not wish to eat nuts because of dental issues. An alternative to eating whole nuts is to incorporate nut butters into the diet. It seems intuitive that nut butters should provide health benefits similar to those seen with whole nuts. However, few studies have investigated whether the benefits of whole nuts can be extrapolated to nut butters. Two recent epidemiological studies (*Luu et al., 2015*; *Van den Brandt & Schouten, 2015*)

reported significantly lower total mortality with increased nut intake, but not nut butter alone. Conversely, the Nurses' Health Study (NHS) showed peanut butter consumption was inversely associated with type 2 diabetes in women (*Jiang et al., 2002*) and in a subset including women with type 2 diabetes, cardiovascular risk was significantly lower among those consuming peanuts and peanut butter at least five times per week (*Li et al., 2009*). Two interventions found significant improvements in blood lipids after consuming different forms of peanuts or almonds (including butter), with no significant differences between groups (*McKiernan et al., 2010*; *Spiller et al., 2003*). While promising, these studies had short intervention periods (28 days) and were possibly underpowered. Longer, adequately-powered studies are required in order to develop firm conclusions. One potentially important difference between nuts and nut butters is the effects on energy balance. As mentioned previously, studies examining body weight (*Alper & Mattes, 2002*; *Fraser et al., 2002*; *Hollis & Mattes, 2007*; *Sabate et al., 2005*; *Tey et al., 2011*) indicate eating whole nuts results in less weight gain than predicted, possibly through increased satiety (*Mattes, 2008*). The effect of nut butters on weight and satiety needs clarification. Also, a recent study reported the measured metabolisable energy of whole almonds (4.42 kcal/g) and almond butter (6.53 kcal/g) differed significantly (*Gebauer et al., 2016*), and thus may have differing effects on energy intake and body weight regulation (*Gebauer et al., 2016*; *Grundy, Lapsley & Ellis, 2016*). Therefore, given there is surprisingly little information on the health benefits of consuming nut butter, it is an important area for future research, so that evidence-based guidelines can be developed.

It is of interest to compare the perceptions of nuts between health professionals and the general public to examine if there are major disparities, which need to be addressed so that there is a consistent understanding of the health benefits of regular nut consumption. For the statements regarding the perceptions of nuts, dietitians were least likely to respond that they "did not know", while the general public were mostly likely to, with the GPs and practice nurses intermediary. There was a similar pattern when analysing agreement and disagreement for the perceptions. Among the top five perceptions to which participants, especially the general public, were most likely to respond "I don't know", were "eating them can increase people's total blood cholesterol", "they are high in antioxidants", "some of them are high in selenium", "some of them are high in iron", and "eating them can help lower people's risk of diabetes". It is not surprising the latter statement received a high prevalence of "I don't know" responses as there is conflicting evidence regarding this in the scientific literature. While one meta-analysis reported a small but significant decrease in the incidence of type 2 diabetes, this study has been criticised on the basis of study selection (one study did not examine the independent effects of nuts), and failure to adjust for BMI (*Afshin et al., 2014*). Conversely, two other meta-analyses have reported no association between nut consumption and the risk of developing type 2 diabetes (*Luo et al., 2014*; *Zhou et al., 2014*). There was also confusion regarding the iron content of nuts. Some nuts can be useful sources of iron, but this will be in the form of non-haeme iron, which is less absorbable and can be influenced by other components of the diets. Also, despite the high prevalence of "I don't know" responses, it is well established that nuts have hypocholesterolaemic properties (*Musa-Veloso et al., 2016*; *Sabate, Oda & Ros, 2010*),

are rich sources of antioxidants (*Bolling et al., 2011*), and that some nuts, in particular, Brazil nuts, are high in selenium (*Thomson et al., 2008*). Interestingly, among those who responded with agreement or disagreement to the perception statements, overall there was greatest agreement that "some of them are high in selenium". This tends to indicate that the message that Brazil nuts are high in selenium is reaching some people. Given that NZ soils are low in selenium and the risk of inadequate intake of selenium has been estimated at 45% (*University of Otago and Ministry of Health, 2011*), Brazil nuts offer a useful source of this nutrient.

The strengths of the study include the recruitment of large national samples and the rigorous survey method using a mixed-mode approach using both an online and paper mail version of the questionnaire to improve response rates.

There are also a number of limitations to be considered when interpreting the data. The cross-sectional nature of the study allows for generation of hypotheses for future research, but means that causal inferences cannot be drawn. The surveys were self-administered, and it is possible that some respondents did not correctly interpret all questions. However, careful development and pre-testing of the questionnaires was undertaken to address this. Response rates were 44% for the general population and 53% for the health professionals. Although these are comparable to other mail surveys in Australasia (*Lee et al., 2005*; *Timperio et al., 2000*), it is possible responders and non-responders differed. While this may influence means, medians, and percentages; there are no clear reasons why the observed associations would differ between responders and non-responders. The results may not be generalisable to countries with healthcare systems that differ from those seen in NZ. Also, we cannot link the perceptions/beliefs of health professionals with their patients due to the design of the study. Linked data would help to determine whether attitudes can be transmitted between these groups. This would provide useful information on whether an intervention, which positively influences health professionals' views on nuts could be expected to influence the views of their patients.

## CONCLUSIONS

It is concerning that so few of the general public report they have received advice from health professionals to consume more nuts, especially given their apparent responsiveness to such advice. Interestingly, both the general public and health professionals consider nuts to be healthy, which may mean that they would be willing to increase intake and advice to increase nuts respectively, if some of the deterrents for eating nuts are addressed. These deterrents included cost, weight gain, and causing one to eat too much fat. We also identified some important motivators to increase nut consumption such as improving health, increasing nutrients in the diet, helping get the right balance of fat, and if they were more affordable. Given that over half of the participants would eat more nuts if advised to by their doctor or dietitian, health professionals should use these potential facilitators to promote regular nut consumption. In addition, these factors should also be addressed in public health messages to encourage regular nut consumption among the public.

## ACKNOWLEDGEMENTS

The authors would like to thank the participants for their commitment and enthusiasm in participating in this study.

### Funding
The authors received no funding for this work.

### Competing Interests
Andrew Robert Gray is an Academic Editor for PeerJ.

### Author Contributions
- Rachel Clare Brown and Andrew Robert Gray conceived and designed the experiments, performed the experiments, analyzed the data, prepared figures and/or tables, authored or reviewed drafts of the paper, approved the final draft.
- Lee Ching Yong conceived and designed the experiments, performed the experiments, analyzed the data, authored or reviewed drafts of the paper, approved the final draft.
- Alex Chisholm and Sook Ling Leong conceived and designed the experiments, performed the experiments, authored or reviewed drafts of the paper, approved the final draft.
- Siew Ling Tey conceived and designed the experiments, authored or reviewed drafts of the paper, approved the final draft.

### Human Ethics
The following information was supplied relating to ethical approvals (i.e., approving body and any reference numbers):

The study was approved by the University of Otago Ethics Committee (reference number D14/288).

### Data Availability
The raw data are provided in Supplemental Information 1.

### Supplemental Information
Supplemental information for this article can be found online at http://dx.doi.org/10.7717/peerj.5500#supplemental-information.

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
