# Peer review of "A comparison of perceptions of nuts between the general public, dietitians, general practitioners, and nurses"

_PeerJ, doi:10.7717/peerj.5500_

## Round 0.1 · original submission · Major Revisions

After considering your Appeal, I am willing to revisit my previous decision. Therefore I am issuing this Revision decision so you can upload your rebuttal and revised manuscript which will be further submitted to peer-review process as well as re-evalauted by myself.

· Appeal

Appeal

· · Academic Editor

Reject

Based on comments from Reviewer 2, indicating that most findings present in this manuscript have already been published, I cannot consider this manuscript as adequate for publication.

·

Basic reporting

No comment

Experimental design

No comment

Validity of the findings

No comment

Additional comments

The manuscript is very well written and Informative. I’d like to thank the authors for all the work they’ve put into it. Some minor comments and suggestions for improvements are below:
INTRODUCTION
Lines 77-84: The suggestion that individuals would increase nut intake if advised by doctor seems to be in contrast to the findings discussed from these studies. Please modify sentences 77 to 78 to reflect the discussion in 78-84.
METHODS
Lines 121, 123: Please state once that the questionnaires used multiple response questions so you don’t need to state it repeatedly.
Lines 142-143: How was this enforced with paper questionnaires?
Lines 156-157: Please state what tests specifically were used. What kind of chi-squared tests, linear regression models?
RESULTS:
General comments: Please show P-value<0.05 whereever significance is mentioned in results section and what effect was significant.
Line 168: Where does the N=1440 come from. It seems like this number should be much lower? Please double check.
Line 200 and other places: Please be consistent with showing the N for all %s.
Lines 205-208: Where is this data shown in Table 3?
Line 216-: Suggest arranging the perception and knowledge of nuts results section so that the strongly agree to strongly disagree results are discussed first and then the ‘I do not know’ results for clarity.
Lines 233-238: Could you show % ranges like shown in the previous paragraph regarding what % agreed, %disagreed?
DISCUSSION
General comments: The results seem to be reiterated in the discussion. The discussion section could be refined further by discussing the implications of the important findings only.
Lines 251, 257: The term ‘significant differences’ with respect to results doesn’t need to be mentioned in the discussion
Lines 265-272: The discussion of the Pawlak et al studies is repetitive. Suggest removing the discussion from the introduction and elaborating it in the discussion only.
Line 364: In addition to Sabate et al 2010, suggest citing a recent review i.e. “The effects of almond consumption on fasting blood lipid levels:a systematic review and meta-analysis of randomised controlled trials”

Reviewer 2 ·

Basic reporting

Good English, appropriate references, well structured and with apparently relevant results

Experimental design

This is not really original research The authors have carried out a moderately large survey amongst health professionals and the general public about beliefs, perceptions and obstacles concerning nut consumption. Unfortunately, they have chosen to slice salamy-style the results and have already published three papers providing somewhat different views of the results. Two of these papers are just cited for the methodology (Brown et al 2017 and Yong et al 2017) and another (Brown RC, Gray AR, Yong LC, Chisholm A, Leong SL, Tey SL. Current nut recommendation practices differ between health professionals in New Zealand. Public Health Nutr. 2018 Apr;21(6):1065-1074. doi: 10.1017/S1368980017003469. Epub 2017 Dec 4.) not cited even though it was published online a while before the present manuscript was sent for publication. Interestingly, these latter paper has the same authors and they are listed in the same order as in the present manuscript.

Thus, this research is far from original and, worse, authors have deliberately omitted citing prior published work that compromises the present one.

Validity of the findings

As per point 2, the findings are invalid.

Additional comments

This article is flawed because most results have already been published, as per the two references cited (brown 2017 and Yong 2017) and another one with exactly the same authors and in the same order has been omitted (Brown RC, Gray AR, Yong LC, Chisholm A, Leong SL, Tey SL. Current nut recommendation practices differ between health professionals in New Zealand. Public Health Nutr. 2018 Apr;21(6):1065-1074. doi: 10.1017/S1368980017003469. Epub 2017 Dec 4). This kind of slicing results of a single investigation is not scientifically sound.

---

## Round 0.2 · Minor Revisions

Please consider the suggestion made by reviewer 3 on an additional paragraph in the Introduction. It is not necessary that you add exactly the one they suggested, but that topic should be mentioned in the Introduction with 2-3 appropriate references.

·

Basic reporting

No comment

Experimental design

No comment

Validity of the findings

No comment

Additional comments

Thank you for revising the manuscript. It is much improved.

Reviewer 3 ·

Basic reporting

no comment

Experimental design

no comment

Validity of the findings

no comment

Additional comments

The presentation of the article is overall good. I feel there is a lack of information regarding other human-health related effects of nuts and the possible molecular mechanisms underlying such effects. Looking at the flow of the paper I believe this part would fit introduction section. Accordingly, please add the following paragraph in the introduction:

Nut consumption has been associated with reduced risk of certain cancers, such as colorectal, endometrial, and pancreatic neoplasms. Evidence regarding nut consumption and neurological or psychiatric disorders is scarce, nonetheless a number of studies indicated significant protective effects against depression, mild cognitive disorders and Alzheimer’s disease. Several molecular mechanisms appear to be implicated in the protective role of nuts, including antioxidant and anti-inflammatory properties, mainly related to their high content in mono- and polyunsaturated fatty acids (PMID: 26586104).

---

## Round 0.3 · accepted · Accept

The authors have included the suggestions provided by the reviewers, so I consider the manuscript is acceptable for publication.